# Policy-Guided Search on Tree-of-Thoughts for Efficient Problem Solving with Bounded Language Model Queries

**Sumedh Pendurkar**  *sumedhpendurkar@tamu.edu*
*Department of Computer Science & Engineering*
*Texas A&M University*

**Guni Sharon**  *guni@tamu.edu*
*Department of Computer Science & Engineering*
*Texas A&M University*

**Reviewed on OpenReview:** *https: // openreview. net/ forum? id=Rlk1bWe2ii*

## Abstract

Recent studies explored integrating state-space search algorithms with *Language Models* (LM) to perform look-ahead on the token generation process, the "Tree-of-Thoughts" (ToT), generated by LMs, thereby improving performance on problem-solving tasks. However, the affiliated search algorithms often overlook the significant computational costs associated with LM inference, particularly in scenarios with constrained computational budgets. Consequently, we address the problem of improving LM performance on problem-solving tasks under limited computational budgets. We demonstrate how the probabilities assigned to thoughts by LMs can serve as a heuristic to guide search within the ToT framework, thereby reducing the number of thought evaluations. Building on this insight, we adapt a heuristic search algorithm, *Levin Tree Search* (LTS), to the ToT framework, which leverages LMs as policies to guide the tree exploration efficiently. We extend the theoretical results of LTS by showing that, for ToT (a pruned tree), LTS guarantees a bound on the number of states expanded, and consequently, on the number of thoughts generated. Additionally, we analyze the sensitivity of this bound to the temperature values commonly used in the final softmax layer of the LM. Empirical evaluation under a fixed LM query budget demonstrates that LTS consistently achieves comparable or higher accuracy than baseline search algorithms within the ToT framework, across three domains (Blocksworld, PrOntoQA, Array Sorting) and four distinct LMs. These findings highlight the efficacy of LTS on ToT, particularly in enabling cost-effective and time-efficient problem-solving, making it well-suited for latency-critical and resource-constrained applications.

## 1 Introduction

*Language Models (LMs)* have demonstrated promising performance across a wide range of natural language processing tasks (Brown et al., 2020; Chowdhery et al., 2023; Achiam et al., 2023). Although LMs are effective on a wide range of language understanding and generation tasks, complex problem-solving tasks such as reasoning tasks remain challenging and often require additional techniques to improve performance. Recent prompting strategies, such as *Chain-of-Thought (CoT) prompting* (Wei et al., 2022), improve performance on such tasks by eliciting intermediate reasoning steps before producing a final response. Building on this idea, methods such as *Tree-of-Thoughts (ToT)* (Yao et al., 2023) (see Figure 1) and *Reasoning-and-Planning (RAP)* (Hao et al., 2023) generate structured reasoning trajectories in the form of search trees. These trees are formed by exploring multiple coherent intermediate sequences of tokens, or *thoughts*, at each decision point. Furthermore, approaches like ToT and RAP employ *state-space search* (referred to as "search") algorithms to navigate the generated search trees and produce a final response. For instance, ToT (Yao et al., 2023) method proposed using *depth first search (DFS)* (Cormen et al., 2022) or *beam search (BS)* (Sutskever et al.,

2014) while RAP (Hao et al., 2023) used *Monte-Carlo Tree Search (MCTS)* (Kocsis & Szepesvári, 2006) as the search algorithm. These approaches have demonstrated significant improvements on complex reasoning tasks compared to standard LM decoding methods that generate responses by greedily selecting tokens without additional exploration.

While these approaches can be effective, the improvements come at the cost of increased inference-time computation. Generating and evaluating multiple reasoning paths requires repeated LM queries, which can be expensive, especially as the size of LMs increases, as highlighted in recent work (Snell et al., 2024; Muennighoff et al., 2025). To address this issue, we focus on improving LM performance under a strictly constrained computational budget, where compute is measured by the number of generated thoughts. Since each thought consists of a sequence of tokens, minimizing the number of thoughts also reduces the total number of LM queries, assuming the average length of each thought remains approximately constant. To this end, we explore adaptations of search algorithms within the ToT framework to elicit better LM responses when the number of allowable thought generations is limited.

We begin with an observation: explicitly evaluating intermediate thoughts using LMs, commonly performed to rank and pick thoughts to expand, requires additional LM queries. Given this understanding, we propose using the LM as a policy, mapping from a partial sequence of thoughts (state) to a distribution over next thoughts (neighboring states), thereby guiding the search without requiring additional evaluation queries. Building on this observation, we adapt *Levin Tree Search (LTS)* (Orseau et al., 2018) to the ToT framework, which directly leverages LM as a policy to effectively guide the search. Further, we extend the theoretical guarantee on the number of state expansions, and consequently on the number of thoughts generated, from the original LTS formulation on generic search trees (Orseau et al., 2018) to the ToT framework. Note that the original LTS theoretical expansion bound does not directly apply to the ToT framework due to its pruned structure. Moreover, we analyze the sensitivity of the bound on the number of thoughts generated, to the temperature parameter, commonly used in the final softmax layer of LMs. Experimental results on three representative domains (Blocksworld, PrOntoQA, Array Sorting) demonstrate that LTS consistently matches or exceeds the accuracy of guided DFS and beam search, search algorithms considered in the original ToT work (Yao et al., 2023), under tight computational budgets. Further, the results show that LTS is competitive or better than guided DFS across three domains and four different LMs. [1]

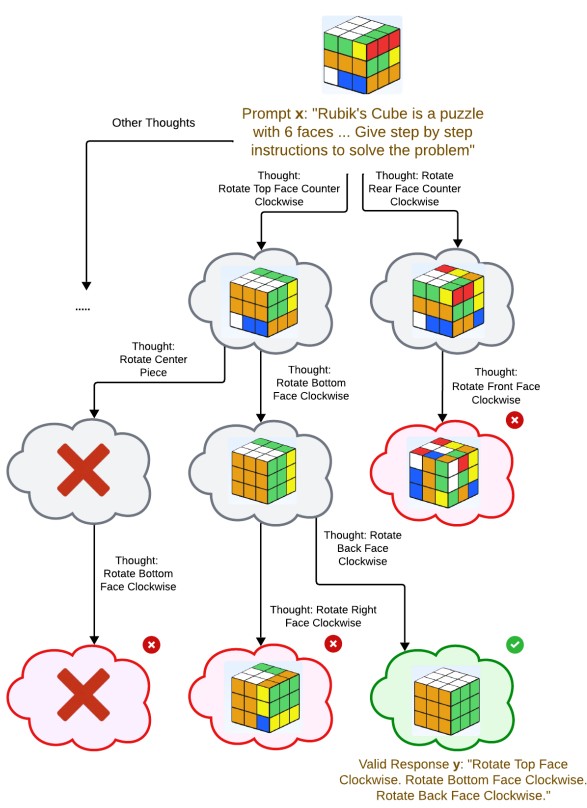

Figure 1: Illustration of the Tree-of-Thoughts (ToT) framework applied to a Rubik's Cube domain. Cube states with red or green backgrounds represent terminal states as determined by the LM, while gray indicates intermediate states. The green background denotes a goal state. States marked with a cross represent invalid states (not known during search) resulting from thoughts that are infeasible for the domain.

---

[1] See https://github.com/sumedhpendurkar/Search-LLM-inference for our code.

## 2 Preliminaries

### 2.1 Problem Formulation

Let $\mathcal{X}$ denote a discrete space of all possible *tokens*, where a token refers to a basic unit of text such as a word, subword, or punctuation mark. The definition of a token is specific to the tokenizer of a given language model (see for example (Grattafiori et al., 2024)). A single token is denoted with a non-bold letter, e.g. $x \in \mathcal{X}$, while a sequence of tokens is denoted with a bold letter, e.g., $\mathbf{x} = (x_1, x_2, \ldots, x_n)$. A *language model (LM)*, $p$, defines a probability distribution over a token given a sequence of previous tokens. Formally, $p : \mathcal{X}^n \mapsto \Delta(\mathcal{X})$, where $\Delta(\mathcal{X})$ is the probability simplex over $\mathcal{X}$ and $n$ is the maximum context length.[2] Given a language model $p$ and an input sequence $\mathbf{x} = (x_1, x_2, \ldots, x_n)$, an *autoregressive decoding method*, $ALG$, generates a response $\mathbf{y} = ALG(p, \mathbf{x})$. For example, this can be done in a greedy way by selecting $p(y_i|\mathbf{x}) = \arg\max p(y_i|y_1, y_2 \ldots y_{i-1}, \mathbf{x})$. In this paper, we assume an access to an evaluator that determines whether a response is valid or not (see Section 4 for details). In our experiments, we consider a fixed sized test set $D = \{\mathbf{x}_1, \mathbf{x}_2, \ldots, \mathbf{x}_n\}$. The decoding methods solve problem-solving tasks, represented by an input sequence $\mathbf{x}_i$, by generating responses $\mathbf{y}_i$ such that $\mathbf{y}_i$ can be considered "valid" by the domain-specific evaluator.

We consider the *Tree-of-Thoughts (ToT)* (Yao et al., 2023) framework for decoding by generating a tree over *thoughts*. Following (Yao et al., 2023), a sequence of tokens $(t_1, t_2, \ldots, t_n)$ is said to be a thought $\mathbf{t}$ if it is a coherent language sequence. The probability of a thought being selected is given by $p(\mathbf{t}|\mathbf{x}) = \prod_{i=1}^{i=n} p(t_i|t_1, t_2 \ldots t_{i-1}, x)$. A sequence of thoughts generated in an autoregressive manner, with the final thought being the end-of-sequence token, constitutes a response $\mathbf{y}$. The original ToT framework suggests two ways of generating next thoughts, (1) sampling – by sampling from the distribution $\mathbf{t}_i \sim p(\mathbf{t}_i|\mathbf{t}_1, \mathbf{t}_2, .., \mathbf{t}_{i-1}, \mathbf{x})$, (2) propose-prompts – using specially designed prompts to generate multiple next thoughts in one go, for example, sequentially listing all actions by appending to output as a part of the LM response. Initially, we consider an alternate strategy, where we consider all possible thoughts that could be generated given $\{\mathbf{t}_1, \mathbf{t}_2, .., \mathbf{t}_{i-1}, \mathbf{x}\}$. This generates a *tree, $\mathcal{T}$*, where each node, referred to as a *state $s \in S$*, is a function of the input query $x$ and the thoughts considered up to that point. The thoughts, $\mathbf{t}$, are edges in $\mathcal{T}$. The neighbor states of $s$ are obtained by appending LM generated thoughts to *s.thoughts*, where *s.thoughts* denotes the sequence of thoughts generated up to state $s$. This process of generating the neighbors (thoughts) is referred to as *expansion*. Note, in $\mathcal{T}$, each state commonly has very high out-degree (referred as *branching factor*). The sampling and propose-prompts based thought generation methods could reduce the branching factor and consequently the search space. We discuss the impact of such methods in Section 3. The descendants of a state in a tree are all states reachable by traversing one or more edges originating from that state. We denote the start state (root node) as $s_0$ where $s_0.thoughts = \phi$. The depth of a state in a tree $\mathcal{T}$ is given by $g(s)$ and we denote depth of the start state as 0, that is, $g(s_0) = 0$. Let $H \subseteq S$ be set of terminal states where the final thought generated consists solely of the end-of-sequence token. A subset of the terminal states $Z \subseteq H$ are valid goal states where $\forall z \in Z$ the final response, *z.thoughts*, is considered valid by the evaluator. The goal of the decoding method is to generate a final response $\mathbf{y}$ by finding a start-to-goal path in tree $\mathcal{T}$. See Figure 1 for an example ToT.

Note, previous methods (Yao et al., 2023) have considered generating multiple start-to-goal paths, and selecting one path (response) by using heuristic driven (LM based) scoring of responses. As we concerned with limited-budget scenarios we only consider the case where the search algorithms return the first solution found as additional calls to LMs are expensive.

### 2.2 Background

Our proposed methods integrates and extends prior work from the LM and heuristic search literature. As such, we discuss the relevant methods.

**Prompt-Based and Search-Based Problem Solving:** Chain-of-Thought (CoT) prompting and its variants (Wei et al., 2022; Kojima et al., 2022) have shown that complex problem solving tasks such as reasoning

---

[2]If the input to a given LM is smaller than $n$ the input is padded.

tasks can be more effectively solved by decomposing the solution process into a sequence of intermediate thoughts. Since CoT selects the next thoughts greedily, it can get stuck in locally optimal paths. Self-Consistency (Wang et al., 2023) mitigates this by sampling multiple paths and selecting the final answer through majority voting. Least-to-Most prompting (Zhou et al., 2023) decomposes the original question into simpler sub-questions, which are then answered sequentially to build up to the final solution. Tree-of-Thoughts (ToT) (Yao et al., 2023) generalizes CoT by treating problem-solving as a tree search over thoughts, allowing backtracking and exploration of multiple paths over the tree using search algorithms, namely, *beam search (BS)* and guided *Depth First Search (DFS)*. Guided Depth-First Search (referred to as DFS henceforth) is a variant of the standard DFS algorithm in which states are evaluated, and the top-ranked state, according to the evaluation function, is selected for expansion. Beam Search is a search algorithm that maintains a fixed number of top-ranked states at each depth, and expands them in parallel. Both DFS and Beam Search use LM to evaluate the states. Reasoning-and-Planning (RAP) (Hao et al., 2023) further extends this line of work by using LMs as reward functions to evaluate partial reasoning chains, for Monte-Carlo Tree Search (MCTS) (Kocsis & Szepesvári, 2006), enabling better performance. Our paper tackles problem-solving tasks similar to those addressed by the Tree-of-Thoughts (ToT) framework, but under a highly limited computational budget. Note, (Zhuang et al., 2024) also address the problem of efficient tree space navigation by leveraging the A* search algorithm (Hart et al., 1968). However, their method assumes access to well-defined cost functions for evaluating states in the search tree. These cost functions are either task-specific or derived from heuristics built on demonstration memory and the LM itself. Thus, they could be hard to obtain for general LM problem-solving tasks. In contrast, our proposed approach relaxes this requirement by operating directly over the probabilistic structure induced by the LM (ToT), avoiding the need for handcrafted or retrieved cost functions.

**LMs for Planning Problems:** Given the recent advancements in LMs (Grattafiori et al., 2024; Achiam et al., 2023) and the integration of ML with planning problems (Orseau et al., 2018; Agostinelli et al., 2019; Pendurkar et al., 2024), recent work has examined integrating LMs and planning solvers. Examples of such integration include (Singh et al., 2023; Ding et al., 2023; Liu et al., 2023; Katz et al., 2024). For instance, (Liu et al., 2023) translates natural language instructions into symbolic planning languages like Planning Domain Description Language (PDDL) (McDermott, 2000) and uses classical planning algorithms. Further, recent work (Koh et al., 2024; Zhou et al., 2024), has explored using LMs to navigate the internet (web automation). However, such methods restrict the environment by enforcing a structure on LM-generated actions (or thoughts) (Liu et al., 2023) or by using a fixed set of actions (Koh et al., 2024) thereby reducing their applicability to domains such as mathematics and logical reasoning. In contrast, our method, built upon the ToT framework, imposes no such constraints, making it broadly applicable to domains where such representations may be infeasible.

**Learning and Search-Based Problem Solving with LMs**: Another line of work also supports the general idea of scaling test-time compute but relies on additional learning (Muennighoff et al., 2025; Li, 2025). For instance, the Policy-Guided Tree Search (PGTS) approach (Li, 2025) investigates using a learned policy to navigate the reasoning search space. However, their method requires additional training of the policy, which can be challenging in settings where supervised signals or reward functions are sparse or expensive to obtain. In contrast, our approach avoids such additional training overhead and remains applicable even in low-resource or inference only scenarios.

### 2.2.1 Levin Tree Search (LTS)

*Levin Tree Search (LTS)* (Orseau et al., 2018) is a policy-guided search algorithm used to solve state-space search problems. LTS uses a policy $\pi : S \mapsto \Delta(S)$, that maps a state $s \in S$ to a distribution over next possible states, as a heuristic to effectively navigate the search space. Note, this mapping can be viewed as a distribution over the *actions* (thoughts) following the original formulation (Orseau et al., 2018). LTS follows a breadth-first search or A* (Hart et al., 1968) like structure where the priority of each state is given by $\min_{s} cost(s)$. Here, $cost(s_i) = \frac{g(s_i)}{\pi(s_i)}$ where $\pi(s_i)$ is defined as the probability of being at state $s_i$ where $\pi(s_i) = \prod_{j=1}^{i} \pi(s_j \mid s_0, s_1, \ldots, s_{j-1})$, where $\{s_0, s_1, \ldots, s_i\}$ is the path from $s_0$ to $s_i$ in the tree. In the context of state-space search literature, a search algorithm is said to expand in best-first order with respect

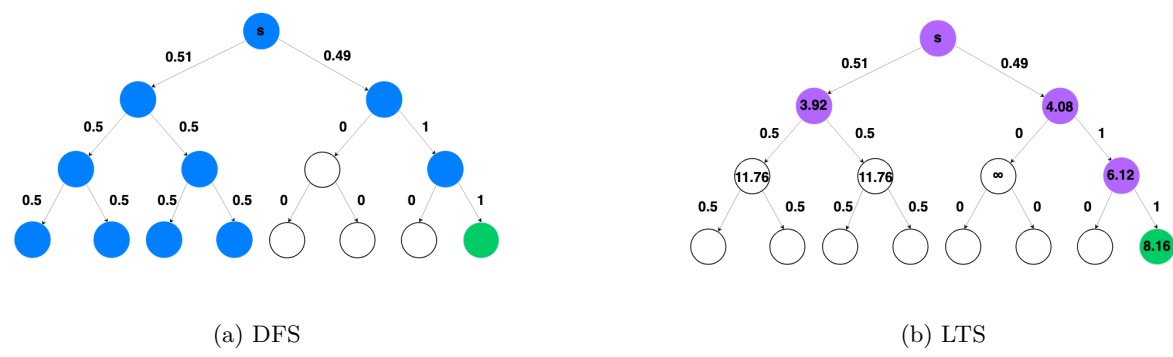

(a) DFS

(b) LTS

Figure 2: Comparison of states expanded by DFS and LTS. State $s$ denotes the starting state and the green state indicates a goal state. The values on edges denote the probability of the next state being selected given the previous state. DFS uses these probabilities as rewards to guide the search. The value inside each state for LTS represents $cost = g(s)/\pi(s)$. The blue and purple states represent the states expanded by DFS and LTS, respectively.

to a cost function $cost(s)$ if, for all states $s_1$ and $s_2$, whenever $cost(s_1) < cost(s_2)$, then $s_1$ is expanded before $s_2$.

The following results from (Orseau et al., 2018, Theorem 2, Theorem 3) hold for any tree.[3]

**Theorem 1.** *LTS expands states in best-first order.*

**Theorem 2.** *Given a tree $\mathcal{T}$ and terminal states $H$, LTS with a policy $\pi$ ensures that the number of state expansions $N$ before reaching any of the terminal states is bounded by*

$$N(\mathcal{T}, H) \leq \min_{s \in H} \frac{g(s)}{\pi(s)}$$

### 2.2.2 Problems with Depth First Search on ToT

In this section, we present a case (with an illustrative example) that suggests DFS, although efficient in its use of LM queries, may not always be the most effective search algorithm for problem-solving tasks. Consider the simple instance depicted in Figure 2 with one goal state (and no additional terminal states) highlighted in green. We restrict our attention to a binary tree, where each state has exactly two successors. Here we assume that the probability values are used as rewards for DFS to select states for expansion. Since DFS follows a greedy strategy, it commits to one (the left, without loss of generality) subtree first and ends up exploring all the states in that subtree. Only after failing to find the goal state does it backtrack and explore the right subtree, where it eventually finds the goal. Whereas for LTS, the successors of the root's left successor each have a *cost* of $3/(0.51*0.5) \approx 11.76$, which is higher than right successor of root which has a *cost* of $\approx 6.12$ and the goal state with $cost \approx 8.16$. As a result, LTS switches to exploring the right subtree, finding the goal state without ever exploring the remainder of the left subtree. Additionally, if we consider the case when all leaf states are terminal states, DFS will return the left most leaf state as the encountered terminal state, which is not a goal state, thereby returning the incorrect solution. Whereas for LTS, the behavior remains unchanged from the previous case, and it will find the goal state (though potentially requiring more exploration than DFS). This case is reflected in our experimental findings (Section 4), where DFS might reach a terminal state more quickly than LTS but often fails to find the goal state, resulting in lower overall accuracy. We refer to this as the *initial-commitment* problem of DFS.

---

[3]While the Theorem 2 holds more generally for graphs with re-expansions, we state it here in the context of trees, which aligns with ToT framework.

## 3 Levin Tree Search with ToT

Guided DFS has two main limitations (1) It requires additional LM queries to determine which state to expand further, and (2) the problem of initial-commitment as discussed in Section 2.2.2. To overcome these limitations, we extend LTS to the ToT framework. We use the LM $p$ as a policy that provides a distribution over possible next thoughts, conditioned on the previously generated token sequence, as described in Section 2.1. In order to adapt to ToT, we propose the following changes. First, given the high vocabulary size of LMs (# tokens) i.e., high branching factor, we use the sampling technique to generate next thoughts $\mathbf{t}_i \sim p(\mathbf{t}_i|\mathbf{t}_1, \mathbf{t}_2, .., \mathbf{t}_{i-1}, \mathbf{x})$, as discussed by Yao et al. (2023). We sample a total of $b_{\max}$ thoughts per expansion. Note that, since we use the LM to guide the search, propose-prompt-based thought generation methods are not applicable to LTS. This is because such methods generate thoughts sequentially (and not independently). As a result, the probability values assigned by the LM during generation are conditioned on previously generated thoughts and, therefore, cannot be directly interpreted as independent heuristic scores. Second, following (Hao et al., 2023), we remove duplicate thoughts generated by sampling. We denote such a subtree of $\mathcal{T}$ as $T$. Lastly, since LTS operates over a tree (rather than a graph), including state-cuts (Orseau et al., 2018) is not meaningful due to the absence of cycles and are, therefore, excluded. The adapted LTS algorithm is presented in Algorithm 1.

---

**Algorithm 1** Levin Tree Search

---

1: **Input:** Query budget $b$, start state $s_0$, LM $p$
2: **Output:** start-to-terminal state path $s_0$ to $h \in H$          ▷ (sequence of tokens/LM response)
3: Initialize $\mathcal{F} \leftarrow \{s_0\}$
4: **while** $\mathcal{F} \neq \phi$ **and** LM queries $\leq$ b **do**
5:      $s \leftarrow \arg\min_{s \in \mathcal{F}} \frac{g(s)}{\pi(s)}$
6:      remove $s$ from $\mathcal{F}$
7:      **if** $s$ is terminal state **then**
8:          **return** path $s_0$ to $s$
9:      **end if**
10:     $A \leftarrow p(\cdot|s).sample()$          ▷ Generate Thoughts
11:     Generate neighbor states $C$ by applying $a \in A$ to $s$
12:     $\mathcal{F} \leftarrow \mathcal{F} \cup C$
13: **end while**

---

Sampling-based methods may expand states in $\mathcal{T}$ in an order that does not follow best-first order. This follows as sampling can potentially prune the best states (with respect to the *cost* function), and thus the state and its descendants might not be expanded. Therefore, Theorem 1 and thus Theorem 2 do not hold (as originally stated) for the ToT framework. Nevertheless, by following a similar line of reasoning, we can derive an analogous bound in which the minimum is taken over the set of terminal states in the sampled subtree $T$, rather than over $H$.

**Proposition 1.** *Given a tree $\mathcal{T}$, $H$ as the terminal states of $\mathcal{T}$, a subtree $T$ generated by sampling, and $H' \subseteq H$ as terminal states in $T$, LTS with a policy $\pi$ ensures that the number of state expansions $N$ before reaching any of the terminal states in $H'$ is bounded by*

$$N(T, H') \leq \min_{s \in H'} \frac{g(s)}{\pi(s)}$$

*Proof.* Let $T_c$ represent the partial search tree consisting only of those states that have been expanded by the time the first terminal state $s_g = \arg\min_{s \in H'} \frac{g(s)}{\pi(s)}$ is expanded during the search process on $T$. Let $\mathcal{L}(T_c)$ denote the leaves of the partial search tree $T_c$. Let $c = \min_{s \in H'} \frac{g(s)}{\pi(s)}$ be the cost of $s_g$. When the first terminal state $s_g$ is expanded, all other states expanded by LTS have a lower cost than $c$. Thus $g(n) \leq \pi(n)c$

for all states $n \in T_c$ expanded by LTS. Therefore,

$$N(T, H') \leq \sum_{n \in \mathcal{L}(T_c)} g(n) \leq \sum_{n \in \mathcal{L}(T_c)} \pi(n)c \leq c = \min_{s \in H'} \frac{g(s)}{\pi(s)}$$

□

Given Proposition 1 and a maximum branching factor $b_{\max}$, the number of thoughts generated can be upper bounded as follows

**Corollary 1.** *Given a tree $\mathcal{T}$, $H$ as the terminal states of $\mathcal{T}$, a subtree $T$ generated by sampling with $b_{\max}$ branching factor, and $H' \subseteq H$ as terminal states in $T$, LTS with a policy $\pi$ ensures that the number of thoughts generated, $M$, before reaching any of the terminal states in $H'$ is bounded by*

$$M(T, H') \leq b_{max} \min_{s \in H'} \frac{g(s)}{\pi(s)}$$

**Note on Renormalization:** It might seem intuitive to re-normalize the sampled thoughts using a method like softmax. However, comparing *cost* of states with different parents (Line 6, Algorithm 1) can be counterintuitive. For example, if the children of a state have a uniform probability distribution, after renormalization, each thought would have a $1/b_{\max}$ probability of being selected. These values are overestimated when compared to other states where probability density is concentrated in top $b_{\max}$ thoughts (sorted with non-increasing probability). This overestimation can mislead the search by making uniformly low confidence thoughts appear more favorable than genuinely high confidence thoughts from other parts of the tree.

## 3.1 Sensitivity to Temperature Parameter

In this section, we analyze the sensitivity of the number of thought generations with respect to the temperature parameter $\tau$ in the final LM softmax layer. We consider a vocabulary of size $V$, where token $y_i$ corresponds to the $i^{\text{th}}$ index in the vocabulary. The probability of selecting $y_i$, given the context (not shown for brevity), is given by

$$p_\tau(y_i) = \frac{e^{\ell_i/\tau}}{\sum_{j=1}^{V} e^{\ell_j/\tau}} \tag{1}$$

where $\ell_1, \ldots, \ell_V$ are the logits of the final layer for a vocabulary of size $V$.

We assume that we have a single goal state $s_g$, and no other terminal states. For a solution path $\{s_0, \ldots, s_g\}$ from $s_0$ to $s_g$, we define a sufficiently accurate LM as

**Definition 1.** *A language model (LM) is said to be **sufficiently accurate** if the following condition holds for all tokens $y^g \in Y^g$, where $Y^g$ is the sequence of tokens selected along the solution path from $s_0$ to $s_g$:*

$$\sum_{j=1}^{V} p_\tau(y_j^g)(\ell_i - \ell_j) \geq \Delta^+$$

*for some non-negative scalar $\Delta^+ \geq 0$, where the $i$-th index corresponds to a token along the solution path.*

**Theorem 3.** *For a sufficiently accurate LM with a fixed parameter $\tau$, let the sampling distribution of tokens be specified by Equation (1). Let each thought consist of at most $k$ tokens and assume that the state $s_g$ achieving the minimum for Corollary 1 is unique. We have*

$$\frac{\partial M(T, H')}{\partial \tau} \leq b_{max} \frac{g(s_g)^2}{\pi(s_g)^2} \cdot \frac{k\Delta^+}{\tau^2}$$

*Proof.* We first consider the partial derivative of $p_\tau(y_i)$.

$$
\begin{aligned}
\frac{\partial p_\tau(y_i)}{\partial \tau} &= \frac{\left(\frac{\partial}{\partial \tau} e^{\ell_i/\tau}\right)\left(\sum_{j=1}^V e^{\ell_j/\tau}\right) - \left(e^{\ell_i/\tau}\right)\left(\frac{\partial}{\partial \tau}\sum_{j=1}^V e^{\ell_j/\tau}\right)}{\left(\sum_{j=1}^V e^{\ell_j/\tau}\right)^2} \\
&= \frac{\left(\frac{-\ell_i}{\tau^2} e^{\ell_i/\tau}\right)\mathcal{S} + e^{\ell_i/\tau}\left(\sum_{j=1}^V \frac{\ell_j}{\tau^2} e^{\ell_j/\tau}\right)}{\mathcal{S}^2} && \left(\text{Let } \mathcal{S} = \sum_{j=1}^V e^{\ell_j/\tau}\right) \\
&= \frac{e^{\ell_i/\tau}}{\tau^2 \mathcal{S}}\left(-\ell_i + \sum_{j=1}^V \ell_j \cdot \frac{e^{\ell_j/\tau}}{\mathcal{S}}\right) \\
&= \frac{p_\tau(y_i)}{\tau^2}\left(-\ell_i + \sum_{j=1}^V \ell_j p_\tau(y_j)\right) && \left(\text{Since } p_\tau(y_i) = e^{\ell_i/\tau}/\mathcal{S}\right) \\
&= \frac{p_\tau(y_i)}{\tau^2}\left(\sum_{j=1}^V p_\tau(y_j)(\ell_j - \ell_i)\right) && \left(\text{Since } \sum_{j=1}^V p_\tau(y_j) = 1, \text{ use } -\ell_i = -\sum_{j=1}^V \ell_i p_\tau(y_j)\right)
\end{aligned}
$$

Next, we consider the derivative for the probability of generating a thought of at most $k$ tokens, $\mathbf{t} = \{y^1, \cdots, y^k\}$. We have $p_\tau(\mathbf{t}) = \prod_{j=1}^k p_\tau(y^j|y^{<j})$.

$$
\begin{aligned}
\frac{\partial p_\tau(\mathbf{t})}{\partial \tau} &= \sum_{j=1}^k \left(\frac{\partial p_\tau(y^j)}{\partial \tau} \cdot \prod_{i \neq j} p_\tau(y^i)\right) && \text{(Product rule)} \\
&\leq \sum_{j=1}^k \left(\frac{\partial p_\tau(y^j)}{\partial \tau}\right) && \left(\prod_{i \neq j} p(w_i) \leq 1\right)
\end{aligned}
$$

Now we analyze the derivative of number of thoughts generated with respect to temperature. Consider a state $s$ at depth $g(s)$ and we have that $\pi(s) = p_\tau(s)$. Each state before this state consists of at most $g(s)$ thoughts, each containing at most $k$ tokens. Since there is a unique path to any state $s$, simply extending the calculation above, we obtain

$$
\frac{\partial \pi(s)}{\partial \tau} \leq \sum_{s_i} \sum_{j=1}^k \left(\frac{\partial p_\tau(y^j)}{\partial \tau}\right) \tag{2}
$$

Note, we ignore indexing notation for the outer summation for brevity.

From Corollary 1, $M(T, H') \leq b_{\max} \min_{s \in H'} \frac{g(s)}{\pi(s)}$. Since we assume that the minimum is achieved at a unique state $s_g$, using Danskin's theorem (Danskin, 1969) we obtain

$$\frac{\partial M(T, H')}{\partial \tau} = b_{\max} \frac{\partial}{\partial \tau} \left( \frac{g(s_g)}{\pi(s_g)} \right)$$

$$= b_{\max} \frac{-g(s_g)}{\pi(s_g)^2} \frac{\partial \pi(s_g)}{\partial \tau}$$

Denoting token along solution path with $y^{g,j}$ with $j^{\text{th}}$ token in the thought

$$\leq -b_{\max} \frac{g(s_g)}{\pi(s_g)^2} \sum_{s_i} \sum_{j=1}^{k} \left( \frac{\partial p_\tau(y^{g,j})}{\partial \tau} \right) \qquad \text{(From Eq 2.)}$$

$$= -b_{\max} \frac{g(s_g)}{\pi(s_g)^2} \sum_{s_i} \sum_{j=1}^{k} \left( \frac{p_\tau(y^{g,j})}{\tau^2} \left( \sum_{a=1}^{V} p_\tau(y_a^{g,j})(\ell_a - \ell_i) \right) \right) \qquad (i \text{ is the optimal token index})$$

$$\leq -b_{\max} \frac{g(s_g)}{\pi(s_g)^2} \sum_{s_i} \sum_{j=1}^{k} \left( \frac{p_\tau(y^{g,j})}{\tau^2} (-\Delta^+) \right) \qquad \text{(From Definition 1)}$$

$$\leq b_{\max} \frac{g(s_g)}{\pi(s_g)^2} \sum_{s_i} \sum_{j=1}^{k} \left( \frac{\Delta^+}{\tau^2} \right)$$

$$= b_{\max} \frac{g(s_g)}{\pi(s_g)^2} \cdot \frac{k \Delta^+ g(s_g)}{\tau^2}$$

$$= b_{\max} \frac{g(s_g)^2}{\pi(s_g)^2} \cdot \frac{k \Delta^+}{\tau^2}$$

$$\square$$

**Remark:** The gradient depends inversely on the square of the probability of state $s_g$, that is, $\pi(s_g)$, and the temperature $\tau$. Based on this dependence, we note that the number of thought generations depend on the values of $\pi(s_g)$ and $\tau$. Increasing $\tau$ causes the probabilities to become more uniform and, in turn, decreases $\pi(s_g)$ when the path from $s_0$ to $s_g$ is the most likely path. If the change is such that the product $\pi(s_g) \cdot \tau$ remains relatively stable, we can expect the number of expansions to be somewhat robust to a change in the temperature. The main inference that can be drawn is that the upper bound is a non-negative number. This suggests that increasing $\tau$ will lead to an increase in the number of LM queries, given that the LM is sufficiently accurate. In experiments we observed that it was often the case that the ToT had multiple goal nodes, as opposed to the theoretical assumption. As such, the theoretical results should be viewed as a general guidance for temperature tuning and not as a strict bound.

## 4 Experiments

This section aims to compare the performance of LTS, DFS, and beam search across multiple domains under varying but constrained LM query budgets.

### 4.1 Setup

This work builds upon the implementation by Hao et al. (2023). The thought generation temperature parameter $\tau$ was set to 0.8. Top-k sampling (Fan et al., 2018) was used with k=50. The maximum branching factor $b_{max}$ for all search algorithms was set to 3. Duplicate thoughts were removed, as duplication is possible during sampling. These hyperparameters were selected following the literature (Hao et al., 2023). For LTS, the temperature for state evaluation (distinct from thought generation) was set to 1 unless stated otherwise. A small value ($\sim 10^{-14}$) was added to the denominator of the *cost* function (to $\pi$) in the LTS algorithm to avoid division by 0. All search algorithms were terminated when either (1) the first terminal state was expanded or (2) a predefined budget was exceeded. The budget is measured as the number of thoughts generated, where each thought is counted twice if additional LM queries are made for state evaluation.

The budget was counted twice for DFS and beam search, as it requires additional LM queries to evaluate the state. This assumes the cost of thought generation is similar to that of state-evaluation. Algorithms were terminated when the first terminal state was found, instead of expanding multiple terminal states and selecting the best, as done in (Hao et al., 2023), to avoid the additional computational overhead from extra LM queries. Our focus is on scenarios with highly limited compute budgets, such as edge-device deployments. Therefore, our experiments and comparisons are restricted to low-budget settings to evaluate performance under tight-computational constraints. We use the Llama 3 instruct models (Grattafiori et al., 2024) (1B, 3B, and 8B), as well as Qwen 2 models (7B) (Team et al., 2024) as LMs in our experiments for two main reasons. First, we aim to evaluate our method under constrained inference settings, and smaller models allow us to rigorously assess test-time compute efficiency in compute-limited scenarios, which is a core focus of this work. Second, Llama and Qwen models are open-weight, high-quality LMs that offer competitive performance (Grattafiori et al., 2024). We note that our method is general and agnostic to the choice of LM, and we expect our findings to extend to other small-scale models beyond Llama models.

## 4.2 Domains

We selected benchmarks following established practices in the ToT literature (Yao et al., 2023; Hao et al., 2023), using PrOntoQA, Blocksworld and a novel Sort as our primary evaluation domains. We did not evaluate all domains from prior work due to practical constraints with small LMs. Specifically, domains such as Game of 24 and Crosswords were excluded because preliminary experiments with Llama-3.2-1B showed high rates of invalid action generation (>40% for Game of 24), where the model produced syntactically incorrect operations or malformed outputs. Since our focus is evaluating search algorithms rather than action validity mechanisms, domains where the base model cannot reliably generate valid actions prevent meaningful algorithmic comparison. StrategyQA was excluded following Yao et al. (2023), who noted that "CoT is already very good on such tasks, and StrategyQA's bottleneck is external knowledge, not reasoning." Creative writing was not included as it involves relatively shallow search trees, limiting our ability to study the effectiveness of search algorithms.

**Blocksworld (BW):** We consider the Blocksworld domain (Valmeekam et al., 2022; 2023), commonly used in heuristic search literature (Pendurkar et al., 2022; 2023), which involves arranging blocks on an infinite table. Given start and goal configurations, the task is to perform a sequence of actions (represented as thoughts) that transition the system from the start to the goal arrangement. The prompt specifies the available actions, which include `stack`, `unstack`, `put`, and `pickup`, each potentially taking block names as arguments depending on the action. A state is said to be terminal if the most recent thought starts with `[PLAN END]` or the max search depth has reached. We follow the same instance grouping strategy as Hao et al. (2023), where BW (step 2) denotes instances solvable with a search depth of 2. We evaluate depths 2, 4, 6, 8, 10, and 12 in our experiments.

**PrOntoQA:** We consider the PrOntoQA (Saparov & He, 2023) domain and follow the same setup as in Hao et al. (2023). Specifically, we adopt the 'true' ontology setting and the 'random' ordering of rules, and merge examples requiring 3 to 5 reasoning hops. A state is terminal if the most recent thought begins with `The answer is`. We use the same 500 sampled instances as Hao et al. (2023), generated using the script provided by Saparov & He (2023). The maximum search depth was set to 10.

**Array Sorting (Sort):** We consider the task of sorting an array with a fixed size of 5 elements, referred to as the Sort domain. The input prompt instructs the LM to perform the sorting using only pairwise swaps, i.e., each thought involves swapping exactly two elements. The array elements may be negative or positive and can include a single decimal point. A terminal state is a state where the most recent thought begins with `Answer:`. The prompt includes instructions so LM is properly instructed. The dataset consists of 100 instances, with the numbers randomly generated. The maximum search depth was set to 10.

## 4.3 Baselines

We consider DFS and beam search as the primary baselines for our experiments. The self-evaluation strategy used to guide DFS and beam search follows Hao et al. (2023) for both the BW and PrOntoQA domains. In these domains, the reward for each generated thought is computed by combining two components: (1) the

log probability assigned by the LM to the thought, and (2) a self-evaluation score derived from a secondary LM prompt that explicitly asks whether the current thought, as an intermediate thought, is valid, using yes/no candidates. For the Sort domain, we rely solely on the LM log probabilities to guide DFS. As a result, each thought is counted as one toward the budget in this setting, since no additional LM queries are required for state evaluation. The beam size used for the beam search algorithm was set to 3, as we observed it offered the best performance for the budgets considered. That is, a higher beam size made beam search sample-inefficient, while a lower beam size acted greedily, similar to DFS.

We exclude MCTS discussed in Hao et al. (2023) due to its known sample inefficiency (Borges & Oliveira, 2021). Furthermore, we do not compare against the approach of Zhuang et al. (2024), which applies the A* search algorithm, because it assumes access to well-defined cost functions that are either handcrafted or derived from demonstrations and heuristics. In contrast, we only consider algorithms that operate directly on the probabilistic structure induced by the LM (ToT), without assuming access to any domain-specific cost functions, as discussed in Section 2.2.1. Consequently, such methods cannot be compared directly to approaches requiring cost functions that are typically unavailable in most domains.

### 4.4 Evaluation

We consider accuracy (success rate) as a metric of evaluation. To determine the correctness of the solution, we assume access to an evaluator, specific to each domain. Note, we do not assume access to an evaluator in real-life scenarios - we select domains where evaluators are accessible to verify the performance of the methods following the literature (Hao et al., 2023; Yao et al., 2023).

**BW:** The generated sequence of thoughts is validated as a plan using a PDDL planner (Valmeekam et al., 2022). If the plan reaches the specified goal configuration, the terminal state is considered a goal state.

**PrOntoQA:** For evaluation, we compare the entire sequence of thoughts against the ground truth answer (with intermediate thoughts) provided in the dataset (Saparov & He, 2023), following the 'entire proof' evaluation proposed by Hao et al. (2023).

**Sort:** The evaluator parses the generated sequence of thoughts to retrieve the sorted array. The evaluator further checks if this array is a sorted version of the input. If the resultant array is sorted, the state is considered a goal state.

### 4.5 Results: Comparison with Increasing LM Query Budget

In this section, we compare the performance of LTS, DFS, and beam search on the ToT framework with increasing budgets. We focus on three representative settings: BW (step 2), BW (step 4), and PrOntoQA (first 100 instances) using the LLaMA 3.2 3B Instruct model. We include BW (step 4) instead of the Sort domain, as the Sort domain typically requires very few thoughts to be generated, making performance trends difficult to observe. The results are presented in Figure 3.

As expected, accuracy increases with the budget and eventually stagnates. The stagnation is clearly seen for all algorithms for BW (step 2), and for DFS for (step 4). Initially, higher budgets allow the algorithm to solve problem instances that were previously unsolved. Once all instances reach their respective terminal states, additional budget does not yield further improvements. For instance, in BW (step 2), accuracy increases until it plateaus at a budget of 15. We observe that LTS generally outperforms DFS across the settings considered, highlighting the advantages of LTS under both low and high budget regimes. The improvement in the high budget regime can be attributed to the issues with DFS outlined in Section 2.2.2, that is, initial-commitment. Further, the improvement in the low budget regime can be attributed to two factors: initial-commitment, and DFS typically requiring additional LM queries for state evaluation. These results suggest that LTS generally performs better than DFS across varying LM query budgets.

An exception occurs at intermediate budget ranges, specifically BW (step 4) at a budget of 15, where DFS slightly outperforms LTS. This can be explained by the greedy nature of DFS, which favors deeper expansions and is more likely to reach a terminal state earlier. If the LM is accurate, DFS may solve instances with fewer state expansions than LTS. It is also worth noting that LTS with a temperature parameter of zero

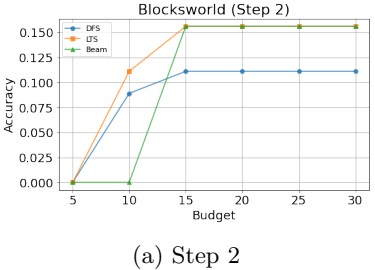 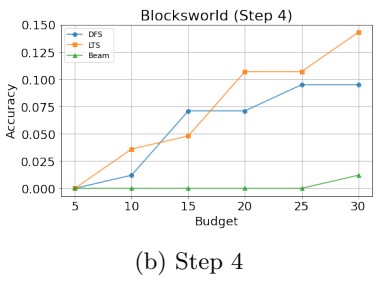 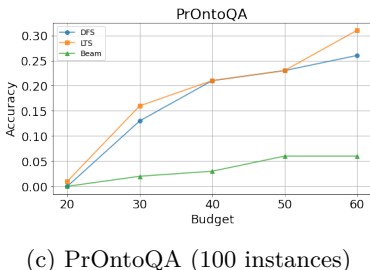

|  (a) Step 2 | (b) Step 4 | (c) PrOntoQA (100 instances) |

Figure 3: Comparison of accuracy of LTS, DFS, and beam search with increasing budgets across 3 different settings.

(used during search) can emulate DFS behavior if a stability term is added at each level and ties in cost are broken in favor of higher $g(s)$ values. LTS can thus offer flexibility across different regimes by interpolating between greedy and balanced exploration of ToT.

On the other hand, when comparing with the performance of beam search, we can see that for BW (step 2) the accuracy is higher than DFS (comparable to LTS) suggesting that exploration helps beam search. However, as 3 states are expanded at each depth, beam search requires a significant budget for search especially with increasing complexity resulting in poor accuracy.

In summary, the results suggest that LTS outperforms or matches DFS across varying LM query budgets. This advantage likely stems from LTS avoiding initial-commitment and incurring fewer LM queries. While beam search performs reasonably well on easier instances (e.g., BW step 2), its performance tends to degrade on harder problems due to sample inefficiency.

## 4.6 Results: Comparison with Increasing Time Budget

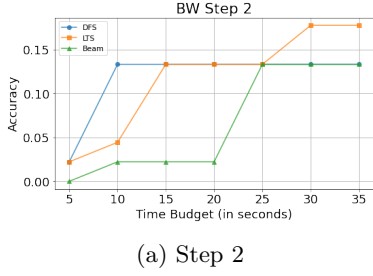 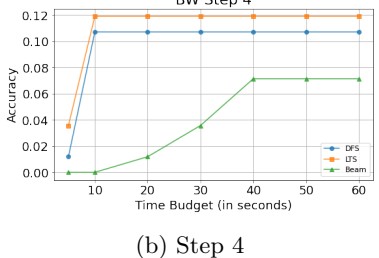 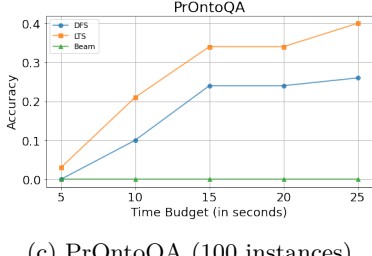

|  (a) Step 2 | (b) Step 4 | (c) PrOntoQA (100 instances) |

Figure 4: Comparison of accuracy of LTS, DFS, and beam search with increasing time budgets across 3 different settings.

In this section, we compare the performance of LTS and DFS using wall-clock time for LM queries (thought generation and evaluation) as the budget metric, rather than the number of LM queries.

Figure 4 presents the accuracy of LTS, DFS, and beam search across three settings: BW (step 2), BW (step 4), and PrOntoQA (first 100 instances) using the LLaMA 3.2 3B Instruct model. The time budget is measured in seconds.

The results show trends similar to those observed with query-based budgets (Section 4.5). LTS generally achieves comparable or higher accuracy than DFS across different time budgets. For BW (step 4) and PrOntoQA, LTS demonstrates consistent improvements over DFS. In BW (step 2), we observe that DFS achieves slightly higher accuracy at certain intermediate time budgets, which aligns with the query-based results and can be attributed to DFS's greedy exploration strategy reaching terminal states more quickly when the LM provides accurate guidance.

Table 1: Accuracy (%) of LTS and DFS across different domain settings and models.

| Task | Model | Size | LTS Accuracy | DFS Accuracy |
|------|-------|------|-------------|-------------|
| BW (step 2) | Llama 3.2 Instruct | 1B | 8.9% | 8.9% |
| BW (step 4) | Llama 3.2 Instruct | 1B | 2.4% | 2.4% |
| BW (step 2) | Llama 3.2 Instruct | 3B | 15.5% | 11.1% |
| BW (step 4) | Llama 3.2 Instruct | 3B | 14.3% | 9.5% |
| BW (step 6) | Llama 3.2 Instruct | 3B | 3.9% | 3.3% |
| BW (step 8) | Llama 3.2 Instruct | 3B | 1.3% | 0.0% |
| BW (step 10) | Llama 3.2 Instruct | 3B | 0.0% | 0.0% |
| BW (step 12) | Llama 3.2 Instruct | 3B | 0.0% | 0.0% |
| BW (step 2) | Llama 3.1 Instruct | 8B | 33.3% | 20.0% |
| BW (step 4) | Llama 3.1 Instruct | 8B | 28.6% | 19.0% |
| BW (step 6) | Llama 3.1 Instruct | 8B | 17.8% | 9.2% |
| BW (step 8) | Llama 3.1 Instruct | 8B | 10.6% | 4.6% |
| BW (step 10) | Llama 3.1 Instruct | 8B | 8.0% | 7.1% |
| BW (step 12) | Llama 3.1 Instruct | 8B | 6.5% | 0.0% |
| BW (step 2) | Qwen 2 | 7B | 31.1% | 26.7% |
| BW (step 4) | Qwen 2 | 7B | 11.9% | 10.7% |
| BW (step 6) | Qwen 2 | 7B | 6.6% | 2.0% |
| BW (step 8) | Qwen 2 | 7B | 2.6% | 2.6% |
| BW (step 10) | Qwen 2 | 7B | 1.8% | 0.9% |
| BW (step 12) | Qwen 2 | 7B | 0.0% | 0.0% |
| Sort | Llama 3.2 Instruct | 1B | 5.0% | 2.0% |
| Sort | Llama 3.2 Instruct | 3B | 20.0% | 15.0% |
| Sort | Llama 3.1 Instruct | 8B | 47.0% | 47.0% |
| Sort | Qwen 2 Instruct | 7B | 26.0% | 22.0% |
| PrOntoQA | Llama 3.2 Instruct | 1B | 18.6% | 15.6% |
| PrOntoQA | Llama 3.2 Instruct | 3B | 27.8% | 18.8% |
| PrOntoQA | Llama 3.1 Instruct | 8B | 70.4% | 49.0% |
| PrOntoQA | Qwen 2 Instruct | 7B | 21.6% | 12.8% |

These findings confirm that the advantages of LTS over DFS extend beyond query efficiency to practical wall-clock time constraints, making it well-suited for latency-critical applications.

### 4.7 Results: Comparison when DFS can reach a terminal state

In this section, we compare the performance of DFS and LTS across all domain settings and the four LMs considered. We do not consider beam search due to its sample inefficiency, as discussed in previous section. The budget for each domain–LM pair was configured such that DFS could reach a terminal state for every instance. In other words, the budget was tuned to maximize DFS performance, but not for LTS, as LTS typically stagnates later than DFS (see Figure 3b). The results are summarized in Table 1. Results for higher-step settings in BW with the 1B and 3B LMs are omitted, as both algorithms achieved 0% accuracy. Overall, LTS performed comparably to or better than DFS in all settings. The largest improvement was observed on PrOntoQA and Llama 3.1 8B, with a 21.4% gain in accuracy. Note, since the proposed method is purely inference-only, its performance is inherently constrained by the quality of the heuristic provided by the underlying LM. Consequently, when the LM heuristic is weak, we do not expect dramatic improvements in absolute accuracy across tasks. These findings suggest that LTS is generally more effective than DFS under constrained budgets and can offer further benefits as the budget increases.

## 4.8 Results: Sensitivity of LTS to Temperature

Figure 5 presents the performance of LTS across different temperature values for the cost function ($\tau \in \{0.01, 0.5, 1.0, 1.5, 2.0\}$) as a function of time budget on Blocksworld Step 4 using Llama 3.2 3B Instruct. Note that the same temperature ($\tau = 0.8$) was used for thought generation across all experiments; the varied temperature parameter applies only to the cost function computation in LTS (Line 6, Algorithm 1), affecting which states are prioritized for expansion.

The results reveal three distinct behavioral regimes. *Low Temperature ($\tau = 0.01$):* At very low temperature, LTS exhibits behavior similar to DFS. The cost function becomes highly sensitive to probability differences, causing the algorithm to strongly commit to high-probability paths. This results in good performance at low budgets (4.5% at 5 seconds), but accuracy plateaus early and reaches only ∼8.5% at 20 seconds. This plateau occurs due to the initial-commitment problem. *Intermediate Temperature ($\tau = 0.5$–$1.5$):* At moderate temperatures, LTS demonstrates improved exploration-exploitation balance. These configurations show relatively poor performance at lower budgets (3.5–3.8% at 5 seconds) but steadily improve as budget increases, ultimately achieving the highest accuracies (10.5–12% at 20 seconds). The initial performance deficit occurs because intermediate temperatures encourage broader exploration

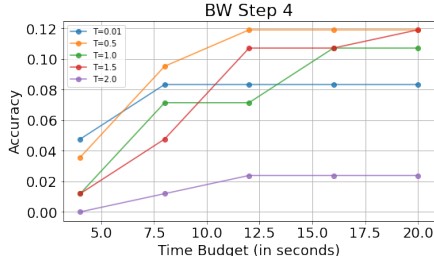

Figure 5: LTS performance with varying temperature $\tau$ values.

rather than immediate commitment to high-probability paths. However, this exploratory behavior enables discovery of superior solutions that low-temperature search misses, resulting in continued accuracy improvements across the entire budget range. *High Temperature ($\tau = 2.0$):* At very high temperature, LTS exhibits breadth-first search (BFS)-like behavior. The flattened probability distribution causes the algorithm to explore states more uniformly, with minimal preference for high-probability paths. This results in the poorest performance at low budgets and modest final accuracy (∼2.1% at 20 seconds). While this temperature could eventually achieve competitive performance, it requires substantially more computational budget to reach accuracy levels that intermediate temperatures achieve more efficiently.

These results suggest that temperature selection involves a trade-off between early performance and ultimate solution quality. In this domain, $\tau \approx 0.5$–$1.0$ offers reasonable early performance while maintaining capacity for continued improvement under budget constraints. When maximizing final accuracy is prioritized over early results, $\tau \approx 0.5$–$1.5$ shows promise, though optimal values may slightly vary based on domain under consideration.

## 5 Summary

This work addresses the challenge of improving Language Model (LM) performance on problem-solving tasks under strict LM query constraints. Specifically, it focuses on the Tree-of-Thoughts (ToT) framework, where a search tree is generated with edges representing sequences of tokens, or thoughts. The paper adapts Levin Tree Search (LTS), a policy-guided search algorithm, to this framework. It also extends the theoretical guarantees of LTS to ToT by bounding the number of state expansions and, consequently, the number of LM generated thoughts. Empirical results suggest that LTS achieves accuracy that is comparable to or better than guided depth first search (DFS) and beam search, which were originally proposed by Yao et al. (2023). Further, the results across three domains and four different LMs show that LTS achieves accuracy that is comparable to or better than guided DFS under tight computational budgets. These findings highlight the effectiveness of LTS within the ToT framework and its applicability to cost-sensitive and time-sensitive settings.

## Acknowledgments

The authors would like to thank Prathamesh Dharangutte for early discussions about this work. The research was conducted in the PiStar AI and Optimization Lab at Texas A&M University. PiStar is supported in part by the NSF (IIS-2238979).

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
