# OpenReview forum: "Policy-Guided Search on Tree-of-Thoughts for Efficient Problem Solving with Bounded Language Model Queries"
_TMLR — Accepted by TMLR_

### Review · Reviewer_Y3EJ · 2025-10-29

**Summary Of Contributions:**

This work presents an algorithm for inference with LLMs using tree search over potential reasoning chains under a fixed and limited compute budget. The authors extend Levin tree search (LTS) to the LLM inference-time search domain by applying it to tree-of-thought, and present a number of theoretical results that bound the compute used by the algorithm as well as its sensitivity to the temperature parameter on LLM sampling. Empirically, the authors evaluate their method on blocksworld, PrOntoQA, and array sorting on 3 sizes of the Llama 3 instruct family. Baselines compared against are DFS and beam search. In general, their method outperforms baselines, especially for lower compute budgets.

**Additional Comments:**

Nit: consider explicitly mentioning the domains you evaluate on in the abstract and intro

**Audience:**

Yes

**Audience Explanation:**

Efficient LLM inference-time search for reasoning is of interest to many in the field.

**Broader Impact Concerns:**

N.A.

**Claims And Evidence:**

No

**Claims Explanation:**

The compute budget is measured by number of thoughts, where each thought is a sequence with a variable number of tokens. Therefore, k thoughts in one tree may not have the same actual compute used as in another method, making this not a good way of measuring compute budgets. Perhaps in the three domains evaluated this is mitigated by the fact that thoughts are likely quite short and structured and similar at each node in the tree, but more generally this will not hold and even so it's unclear now how much of an issue this is. Moreover, assuming cost of generation is the same as cost of evaluation is also not true, as evaluation of a sequence can be done in parallel over the sequence dimension, whereas generation has to be done autoregressively.

Only the llama models are used for experiments, but these models behave differently from other models like Qwen as shown in this paper https://arxiv.org/abs/2503.01307. At expect the results when proposing a general method for inference to be validated on at least on other model family such as Qwen or Olmo.

**Requested Changes:**

Required for acceptance:
- Change the compute budget measure to actually measure compute used instead of the something that varies per method
- Evaluate on another model family

Would strengthen the work:
- Ablate the aspects of the method that make it work
- Present results for multiple hyperparameter settings
- Compare against other SotA inference-time search methods
- Evaluate on reasoning models, which are the main models used for reasoning on the types of tasks used in this paper.

---

> ### Author Response · Authors · 2025-11-26
> **Authors Response to Review by Reviewer Y3EJ**
>
> For required for acceptance changes - We have made the requested changes to the paper. Please see our responses to common concerns 1 and 2 for details.
>
> - Ablate the aspects of the method that make it work. Present results for multiple hyperparameter settings
>
> **Response:** For clarity: $\tau$ (the temperature parameter governing LTS search) is the sole hyperparameter our approach introduces, controlling how the search balances exploration versus exploitation. Other parameters such as branching factor b and LM sampling temperature are inherited from existing work (Yao et al. 2023; Hao et al. 2023) and impact all compared methods uniformly.
> Concerning temperature sensitivity analysis: Our choice of temperature=0.8 stems from initial experiments indicating it provided an effective balance between exploring new paths and exploiting promising ones. Please refer to common concern 3 for results on sensitivity analysis.
>
> - Compare against other SotA inference-time search methods
>
> **Response:** We are unaware of other SotA inference-time search methods which align with our problem definition. We have discussed and considered other methods (see Section 4.3) however, they cannot be directly applied without significant changes under the problem formulation considered in the paper. This is highlighted in our paper in Section 4.3, and we have revised the text for clarification “We exclude MCTS discussed in Hao et al. (2023)  due to its known sample inefficiency (Borges & Oliveira, 2021). Furthermore, we do not compare against the approach (Zhuang et al. (2024)) which applies the A* search algorithm, because it assumes access to well-defined cost functions that are either handcrafted or derived from demonstrations and heuristics. In contrast, we only consider algorithms that operate directly on the probabilistic structure induced by the LM (ToT), without assuming access to any domain-specific cost functions, as discussed in Section 2.2.1. Consequently, such methods cannot be compared directly to approaches requiring cost functions that are typically unavailable in most domains.”
>
> - Evaluate on reasoning models, which are the main models used for reasoning on the types of tasks used in this paper.
>
> **Response:** We only choose small models as, in this paper, we are concerned with LM inference on edge devices where compute is limited and using large models would not be feasible. We have revised the text in Section 4.1 “smaller models allow us to rigorously assess test-time compute efficiency in compute-limited scenarios, which is a core focus of this work” for more clarity.
>
> - consider explicitly mentioning the domains you evaluate on in the abstract and intro:
>
> **Response:** Sure - we have added it.
>  Abstract: “Empirical evaluation under a fixed LM query budget demonstrates that LTS consistently achieves comparable or higher accuracy than baseline search algorithms within the ToT framework, across three domains (Blocksworld, PrOntoQA, Array Sorting) and four distinct LMs”. Introduction: "Experimental results on three representative domains (Blocksworld, PrOntoQA, Array Sorting) demonstrate that LTS consistently matches or exceeds the accuracy of guided DFS and beam search"

---

> > ### Comment · Reviewer_Y3EJ · 2025-11-28
> > **Thanks for the response**
> >
> > Quick clarifying question, while going over the rebuttal I noticed you didn't evaluate Qwen on the sorting task, is that correct? Why is that?

---

> > > ### Author Response · Authors · 2025-11-30
> > > **Response to Reviewer Y3EJ's comment**
> > >
> > > Thank you for catching this omission, and we apologize for this oversight. We have now completed these experiments and the results are as follows: with Qwen LTS achieves 26% accuracy and DFS achieves 22% on the sorting task.
> > >
> > > These results align with the consistent patterns we observed across (1) Llama's performance on the sorting task, and (2) Qwen's behavior on the other evaluation domains. Specifically, Qwen demonstrates similar performance characteristics to what we saw with Llama on sorting, further validating the generalizability of our findings across different LMs. We have updated the paper accordingly.

---

### Review · Reviewer_sEzZ · 2025-10-31

**Summary Of Contributions:**

The paper adapts Levin Tree Search (LTS) to the Tree-of-Thoughts (ToT) setting, using LM token/thought probabilities as a policy to prioritize expansions without additional self-evaluation queries; it extends LTS guarantees to pruned, sampling-induced ToT subtrees (Proposition 1, Corollary 1) and analyzes sensitivity to softmax temperature (Theorem 3). 2) Empirically, under fixed query budgets across Blocksworld (multiple depths), PrOntoQA, and a 5-element array-sorting task with Llama-3 (1B/3B/8B), LTS generally matches or exceeds guided DFS and is more budget-efficient than beam search; results include accuracy vs. budget curves and a table tuned so DFS can reach a terminal state. Strengths: principled policy-guided search with budget awareness; a clear theoretical bound adapted to ToT; simple, training-free drop-in replacement for DFS; consistent gains on harder settings. Weaknesses: budgets are counted asymmetrically (DFS incurs double counting when self-evaluating), potentially biasing comparisons; wall-clock/tokencost latency is not reported; accuracy levels on several BW depths remain low; evaluation stops at the first terminal (not necessarily goal) state; experiments exclude stronger/alternative planners (e.g., A*-style ToT variants) and larger LMs; the “sufficiently accurate LM” assumption and uniqueness conditions limit the generality of the temperature analysis.

**Audience:**

Yes

**Audience Explanation:**

Readers working on controllable inference costs, reasoning with small/edge LMs, or routing/planning for agents will find actionable ideas and a baseline worth adding to their stacks. 2) While the scope is limited (small models, synthetic/controlled tasks), the method’s simplicity, theoretical clarity, and noted wins over DFS under tight budgets make it a useful reference and a plausible component in future ToT toolkits.

**Broader Impact Concerns:**

Policy-guided search that favors high-probability continuations risks entrenching LM prior biases (e.g., stereotyping or unsafe defaults) when used in user-facing agents; budget-driven pruning may systematically ignore low-probability but correct branches in safety-critical contexts. 2) If deployed to reduce inference cost at scale, selection pressure toward “cheap but confident” reasoning could degrade deliberative quality unless paired with risk controls; the paper should add guidance on domains requiring fail-safes, human-in-the-loop escalation, and audits for bias amplification under policy-guided pruning.

**Claims And Evidence:**

Yes

**Claims Explanation:**

The adaptation of LTS to ToT is clearly stated with Algorithm 1 and a bound replacing \min_{s\in H} by \min_{s\in H’} over sampled subtrees; the DFS failure mode is illustrated, and the temperature analysis supplies an interpretable comparative-statics result. 2) Yet the experiments count DFS thoughts twice when using self-evaluation, terminate upon the first terminal rather than the best terminal, and omit latency/token-cost reporting—each choice can tilt the cost-efficiency comparison; accuracy remains modest for many BW depths, and beam search is dismissed as “sample-inefficient” without equalized compute tuning across tasks. 3) The domains (BW, PrOntoQA, toy sorting) and small LMs are a reasonable starting point but limit external validity; ablations on b_{\max}, k, temperature, and sampling randomness are lite, and comparisons to A*-style ToT methods are only discussed, not executed. Collectively, these choices support the core idea but fall short of a fully convincing case across settings.

**Requested Changes:**

1. Fairness and Budget Accounting (Critical): The paper’s comparison between LTS and DFS is weakened by inconsistent budget definitions—DFS is penalized by double-counting self-evaluations while LTS is not. The authors should re-run experiments with matched. 2. Stopping Criterion Bias (Critical): Current evaluations stop at the first terminal state rather than the best terminal state, unfairly disadvantaging depth-first methods. The authors should modify this to “best-found terminal within budget” or report both variants, clarifying how this affects success rates and search efficiency. 3.The experiments cover only synthetic or simplified reasoning tasks (Blocksworld, PrOntoQA, array sorting) and small models (≤8B). Adding at least one real-world or tool-assisted reasoning benchmark, and testing on a stronger LM or open-weight alternative, would significantly strengthen the empirical claims and generality of findings.

---

> ### Author Response · Authors · 2025-11-26
> **Authors Response to Reviewer sEzZ [1/2]**
>
> - Fairness and Budget Accounting (Critical): The paper’s comparison between LTS and DFS is weakened by inconsistent budget definitions—DFS is penalized by double-counting self-evaluations while LTS is not. The authors should re-run experiments with matched.
>
> **Response:** We have added additional subsection based on the suggestion. Please see common concern 1 for details.
>
> - Stopping Criterion Bias (Critical): Current evaluations stop at the first terminal state rather than the best terminal state, unfairly disadvantaging depth-first methods. The authors should modify this to “best-found terminal within budget” or report both variants, clarifying how this affects success rates and search efficiency.
>
> **Response:** In our paper we consider a budget constraint scenario where we would return the first terminal node (goal node as believed by LM giving the End of Answer token). Accumulating multiple terminal nodes (referred to as anytime algorithm [1]), introduces additional complexity of deciding which terminal node to return, and the performance could be impacted by the choice of prompts used by LMs. As our goal in this paper was to study the effectiveness of (adapted) LTS on tree-of-thoughts, we focus on the setting of returning the first terminal node. We have clarified this setup detail in section  2.1 by adding the text “Note that prior work (Yao et al. 2023) has explored generating multiple start-to-goal paths and then selecting a final response using heuristic, LM-based scoring. However, because we focus on limited-budget scenarios, we restrict our setting to search algorithms that return the first solution found, as additional LM calls are costly. Moreover, selecting among multiple candidate paths introduces additional complexity, as the scoring itself requires extra LM prompts whose reliability and cost must also be managed. ” We also refer the reviewer to cases like in Section 2.2.2 where DFS would naively back-up only one depth regardless of the heuristic scores, on the other hand LTS would explore the right node of root considering the heuristic scores.
>
> [1] Hendler, James, ed. Artificial Intelligence Planning Systems: Proceedings of the First Conference (AIPS 92). Elsevier, 2014.
>
> - beam search is dismissed as “sample-inefficient” without equalized compute tuning across tasks.
>
> **Response:** As stated in the paper, we have indeed tuned the beam size = 3 across domains as that resulted in best performance. See “The beam size used for the beam search algorithm was set to 3, as we observed it offered the best performance for the budgets considered. That is, a higher beam size made beam search sample-inefficient, while a lower beam size acted greedily, similar to DFS.”
>
> - comparisons to A*-style ToT methods are only discussed, not executed.
>
> **Response:** We have considered other methods (see Section 4.3). However, they (including A* style methods) cannot be directly applied without significant changes under the problem formulation considered in the paper. This is highlighted in our paper in Section 4.3, and we have revised the text for clarification “We exclude MCTS discussed in Hao et al. (2023)  due to its known sample inefficiency (Borges & Oliveira, 2021). Furthermore, we do not compare against the approach (Zhuang et al. (2024)) which applies the A* search algorithm, because it assumes access to well-defined cost functions that are either handcrafted or derived from demonstrations and heuristics. In contrast, we only consider algorithms that operate directly on the probabilistic structure induced by the LM (ToT), without assuming access to any domain-specific cost functions, as discussed in Section 2.2.1. Consequently, such methods cannot be compared directly to approaches requiring cost functions that are typically unavailable in most domains.”

---

> > ### Author Response · Authors · 2025-11-26
> > **Authors Response to Reviewer sEzZ [2/2]**
> >
> > - The experiments cover only synthetic or simplified reasoning tasks (Blocksworld, PrOntoQA, array sorting) and small models (≤8B). Adding at least one real-world or tool-assisted reasoning benchmark, and testing on a stronger LM or open-weight alternative, would significantly strengthen the empirical claims and generality of findings.
> >
> > **Response:** Given the concern on only using Llama models we provide additional results on another open source alternative Qwen 2 7B. The results follow a similar trend as for Llama, that is, LTS is on-par or better than DFS when DFS can solve all instances. We only choose small models as, in this paper, we are concerned with LM inference on edge devices where compute is limited and using large models would not be feasible. We have revised the text in Section 4.1 “smaller models allow us to rigorously assess test-time compute efficiency in compute-limited scenarios, which is a core focus of this work” as well as appended the results on Qwen 2 7B in Table 1.
> >
> > We selected benchmarks following established practices in the ToT literature while accounting for the capabilities of small LMs (3B parameters). Our domain choices are grounded in empirical observations. We use ProntoQA and Blocksworld, which are standard benchmarks used by Hao et al. (2023) and Yao et al. (2023). These domains provide appropriate complexity for evaluating search algorithms with small models. Following are the reasons on why some domains are excluded:
> > StrategyQA: As noted by Yao et al. (2023), "CoT is already very good on such tasks, and StrategyQA's bottleneck is external knowledge, not reasoning," making it unsuitable for evaluating search-based reasoning improvements.
> > Game of 24 and Crosswords: We did not consider the Game of 24 or crossword domain as the actions generated for LMs were invalid, resulting in several invalid nodes when tested with smaller models. Since our focus is on evaluating search algorithms rather than action validity checking, domains where the base model cannot reliably generate valid actions prevent meaningful algorithmic comparison.
> > Creative Writing: The original ToT work on creative writing involved relatively shallow search trees, limiting the ability to study the effectiveness of search algorithms.
> > We add the following clarification to Section 4.2: "We selected benchmarks following established practices in the ToT literature Yao et. al 2023, Hao et al. 2023, using ProntoQA, Blocksworld and a novel Sort  as our primary evaluation domains. We did not evaluate all domains from prior work due to practical constraints with small LMs. Specifically, domains such as Game of 24 and Crosswords were excluded because preliminary experiments with Llama-3.2-1B showed high rates of invalid action generation (>40\% for Game of 24), where the model produced syntactically incorrect operations or malformed outputs. Since our focus is evaluating search algorithms rather than action validity mechanisms, domains where the base model cannot reliably generate valid actions prevent meaningful algorithmic comparison. StrategyQA was excluded following Yao et al. 2023, who noted that ``CoT is already very good on such tasks, and StrategyQA's bottleneck is external knowledge, not reasoning.'' Creative writing was not included as it involves relatively shallow search trees,  limiting our ability to study the effectiveness of search algorithms."

---

### Review · Reviewer_GpBr · 2025-11-04

**Summary Of Contributions:**

**Summary of Contributions**: This paper proposes Levin Tree Search (LTS) as a principled alternative to standard search strategies such as DFS and Beam Search in Tree-of-Thought (ToT) reasoning with large language models.  The authors adapt theoretical guarantees from LTS to demonstrate bounded expansion complexity and report minor yet consistent improvements over DFS on three symbolic reasoning benchmarks (BlocksWorld, PrOntoQA, and Sort). Overall, the paper provides a clear theoretical framework of ToT as policy-guided search, although its empirical scope and practical validation remain limited.

**Strengths**
- [**S1**] - **Motivation**: This paper presents a timely and well-motivated approach to decoding in the context of LMs for reasoning tasks.  Overall, the idea of leveraging more sophisticated tree-search algorithms, especially within the limited budget regimes, is excellent.
- [**S2**] - **Computation Results**: Overall, there is a relatively consistent trend in improvements over DFS and beam search, so there is some promise to the computational results.  That said, there are some limitations that I will discuss in the weaknesses.

**Weaknesses**
- [**W1**] - **Experimental Depth**: The empirical evaluation is limited to three toy benchmarks (BlocksWorld, PrOntoQA, Sort).  While these tasks provide reasonable environments for algorithmic comparison, they do not convincingly demonstrate the method’s generality or robustness in more complex reasoning tasks. The absence of larger, semantically rich benchmarks (e.g., StrategyQA) makes it difficult to assess whether the observed improvements of Levin Tree Search (LTS) over DFS or beam search would hold under more realistic reasoning settings.  Additionally, it would be beneficial to include comparisons with non-budget-limited methods to demonstrate the trade-off in terms of time/budget/quality.  Moreover, with only three tasks and very marginal gains over DFS, I do not find the results strongly convincing.  In addition, the authors consistently report results within a limited budget of prompts; however, including the times would also be informative.
- [**W2**] - **Ablation**: The paper does not include essential ablation analyses for hyperparameters that strongly influence search behavior and computational efficiency.  Specifically, the branching factor $b_{\max}$, the temperature $\tau$ in the LM sampling policy, and the top-$k$ sampling cutoff. Each of these parameters directly affects the trade-off between exploration and exploitation and thereby the total query cost. For example, Section 3.1 presents a theoretical dependence of search cost on $\tau$, but this relationship is never empirically validated. Similarly, the choices of $b_{\max}$ and $k$ are fixed without justification or sensitivity analysis, leaving unclear how significantly these parameters affect the trade-off between solution quality and computational burden.  Including these ablations would substantially strengthen the motivation for these parameters, while also providing practical guidance on the trade-offs associated with the proposed method.
- [**W3**] - **Clarity**: I find this submission to suffer from a lack of clarity.  The abstract is overly jargon-heavy.  The theory is presented with little to no explanation of how these choices impact practicality in this setting, and there is even less empirical justification for it.  While the authors illustrate a DFS failure case, similar counterexamples likely exist for the proposed method, suggesting an unavoidable no-free-lunch trade-off in exploration efficiency.  As such, I don't find this section to be compelling.  Additionally, a brief section on the specific contributions should be included.

**Additional Comments:**

I do not research LMs and have relatively limited experience.  I believe my weaknesses and suggestions for improvement are reasonable, regardless of this.  However, having a reviewer more familiar with the literature, existing baselines, and benchmarks would be helpful in thoroughly assessing the work.

**Audience:**

Yes

**Audience Explanation:**

This paper presents a a policy-guided approach for bounded LM queries, making it a good fit for TMLR.

**Broader Impact Concerns:**

None.

**Claims And Evidence:**

No

**Claims Explanation:**

Overall, most of the claims have some evidence.  However, as pointed out in **W1/2** some claims could be strengthened with more robust experimentation.

**Requested Changes:**

Overall, the authors definiteily need to include ablation on key parameters of their approach **W2** and improve the clarity of the writing **W3**.  Additionally, improvement in the direction of **W1** would strengthen the work.

---

> ### Author Response · Authors · 2025-11-26
> **Authors Response to Reviewer GpBr [1/n]**
>
> **W1:** The empirical evaluation is limited to three toy benchmarks (BlocksWorld, PrOntoQA, Sort). While these tasks provide reasonable environments for algorithmic comparison, they do not convincingly demonstrate the method’s generality or robustness in more complex reasoning tasks. The absence of larger, semantically rich benchmarks (e.g., StrategyQA) makes it difficult to assess whether the observed improvements of Levin Tree Search (LTS) over DFS or beam search would hold under more realistic reasoning settings.
>
> **Response:**
> We selected benchmarks following established practices in the ToT literature while accounting for the capabilities of small LMs (3B parameters). Our domain choices are grounded in empirical observations. We use ProntoQA and Blocksworld (and a novel Sort), which are standard benchmarks used by Hao et al. (2023) and Yao et al. (2023). These domains provide appropriate complexity for evaluating search algorithms with small LMs. Following are the reasons on why some domains are excluded:
> StrategyQA: As noted by Yao et al. (2023), "CoT is already very good on such tasks, and StrategyQA's bottleneck is external knowledge, not reasoning," making it unsuitable for evaluating search-based reasoning improvements.
>  Game of 24 and Crosswords: We did not consider the game of 24 or crossword domain as the actions generated for LMs were invalid resulting in several invalid nodes when tested with smaller models. Since our focus is on evaluating search algorithms rather than action validity checking, domains where the base model cannot reliably generate valid actions prevent meaningful algorithmic comparison.
> Creative Writing: Yao et al. 2023 presented results on the ‘creative writing’ domain which has relatively shallow search trees, limiting the ability to study the effectiveness of search algorithms.
> We add the following clarification to Section 4.2: "We selected benchmarks following established practices in the ToT literature (Yao et. al 2023, Hao et al. 2023), using ProntoQA, Blocksworld and a novel Sort  as our primary evaluation domains. We did not evaluate all domains from prior work due to practical constraints with small LMs. Specifically, domains such as Game of 24 and Crosswords were excluded because preliminary experiments with Llama-3.2-1B showed high rates of invalid action generation (>40% for Game of 24), where the model produced syntactically incorrect operations or malformed outputs. Since our focus is evaluating search algorithms rather than action validity mechanisms, domains where the base model cannot reliably generate valid actions prevent meaningful algorithmic comparison. StrategyQA was excluded following Yao et al. 2023, who noted that ``CoT is already very good on such tasks, and StrategyQA's bottleneck is external knowledge, not reasoning.'' Creative writing was not included as it involves relatively shallow search trees, limiting our ability to study the effectiveness of search algorithms."

---

> ### Author Response · Authors · 2025-11-26
> **Authors Response to Reviewer GpBr [2/n]**
>
> **W1:** Additionally, it would be beneficial to include comparisons with non-budget-limited methods to demonstrate the trade-off in terms of time/budget/quality.
>
> **Response:** The motivation of our paper lies when budget is highly limited, for example on edge devices, and thus we only demonstrate the results on such low budget settings and considering methods that are well suited in such cases. We have added text in Section 4.1 “Our focus is on scenarios with highly limited compute budgets, such as edge-device deployments. Therefore, our experiments and comparisons are restricted to low-budget settings to evaluate performance under tight-computational constraints.”
>
> **W1:** Moreover, with only three tasks and very marginal gains over DFS, I do not find the results strongly convincing.
>
> **Response:** We expect LTS to explore the ToT more efficiently by balancing exploration and exploitation. For example, Section 2.2.2 illustrates concrete cases where DFS can fail due to its overly myopic exploration strategy, while LTS is able to recover by allocating search more adaptively. However, since our method is purely inference-only, its performance is inherently constrained by the quality of the heuristic signal provided by the underlying LM. Consequently, when the LM heuristic is weak, we do not expect dramatic improvements in absolute accuracy across tasks.
> Despite this, LTS still delivers meaningful gains: on PrOntoQA with Qwen2-7B, LTS solves approximately 1.7 times more problem instances correctly, which we view as a significant improvement given the low-budget, inference-only setting.
> We have updated text in Section 4.6 “Note, since the proposed method is purely inference-only, its performance is inherently constrained by the quality of the heuristic provided by the underlying LM. Consequently, when the LM heuristic is weak, we do not expect dramatic improvements in absolute accuracy across tasks.”
>
> **W1:** In addition, the authors consistently report results within a limited budget of prompts; however, including the times would also be informative.
>
> **Response:** We have added additional experiments. Please see our results in common concern 1.
>
> **W2**: The paper does not include essential ablation analyses for hyperparameters that strongly influence search behavior and computational efficiency. Specifically, the branching factor, the temperature in the LM sampling policy, and the top- sampling cutoff. Each of these parameters directly affects the trade-off between exploration and exploitation and thereby the total query cost. For example, Section 3.1 presents a theoretical dependence of search cost on, but this relationship is never empirically validated. Similarly, the choices of  and  are fixed without justification or sensitivity analysis, leaving unclear how significantly these parameters affect the trade-off between solution quality and computational burden. Including these ablations would substantially strengthen the motivation for these parameters, while also providing practical guidance on the trade-offs associated with the proposed method.
>
> **Response:** To clarify: the only hyperparameter introduced by our method is $\tau$ (temperature for LTS search), which controls the exploration-exploitation tradeoff. All other hyperparameters (branching factor b, LM sampling temperature) follow prior work (Yao et al. 2023; Hao et al. 2023) and affect all baseline methods equally.
> Regarding temperature for LTS: We selected temperature=0.8 based on preliminary experiments showing it balanced exploration and exploitation effectively. For sensitivity analysis on $\tau$ please refer to the newly added results discussed in common concern 3.

---

> > ### Author Response · Authors · 2025-11-26
> > **Authors Response to Reviewer GpBr [3/n]**
> >
> > **W3:** I find this submission to suffer from a lack of clarity. The abstract is overly jargon-heavy.
> >
> > **Response:** We have revised our abstract for more clarity. The changes are as follows "to perform look-ahead on the "Tree-of-Thoughts" (ToT) generated by LMs," -> "to perform look-ahead on the token generation process, which is known as the ``Tree-of-Thoughts'' (ToT), generated by LMs,". "we adapt Levin Tree Search (LTS) to the ToT framework," -> "we adapt a state of the art heuristic search algorithm, denoted Levin Tree Search (LTS), to the ToT framework,"
> >
> > **W3:** The theory is presented with little to no explanation of how these choices impact practicality in this setting, and there is even less empirical justification for it.
> >
> > **Response:** For the effects please see our updated remark in Section 3.1 “The gradient depends inversely on the square of the probability of state $s_g$, that is, $\pi(s_g)$, and the temperature $\tau$. Based on this dependence, we note that the number of thought generations depend on the values of $\pi(s_g)$ and $\tau$. Increasing $\tau$ causes the probabilities to become more uniform and, in turn, decreases $\pi(s_g)$ when the path from $s_0$ to $s_g$ is not the most likely path. If the change is such that the product $\pi(s_g) \cdot \tau$ remains relatively stable, we can expect the number of expansions to be somewhat robust to a change in the temperature. The main inference that can be drawn is that the upper bound is a non-negative number.  This suggests that increasing $\tau$ will lead to an increase in the number of LM queries, given that the LM is sufficiently accurate.” In experiments we observed that it was often the case that the tree-of-thoughts had multiple goal nodes, as opposed to the theoretical assumption. As such, the theoretical results should be viewed as a general guidance for temperature tuning and not as a strict bound. For discussion on how these results impact selection for experiments please refer to our answer in common concern 3.
> >
> > **W3:** While the authors illustrate a DFS failure case, similar counterexamples likely exist for the proposed method, suggesting an unavoidable no-free-lunch trade-off in exploration efficiency. As such, I don't find this section to be compelling. Additionally, a brief section on the specific contributions should be included
> >
> > **Response:** The DFS case is illustrated to highlight the failure cases with DFS, and demonstrate the need to explore more in order to achieve a suitable solution. As such, we present LTS which can balance both exploration and exploitation effectively (better exploration than DFS and better exploitation than Beam Search/BFS). Note, the performance of LTS (and most of the heuristic search algorithms) is indeed limited by the quality of heuristics and we do expect LTS to not perform well in cases where the quality of heuristic is poor.

---

### Author Response · Authors · 2025-11-26
**Common Concerns [1/2]**

We appreciate the reviewers' efforts and their assessment of our work. Below, we list the common concerns raised and provide responses to individual concerns in our replies to each review. Please see our responses and let us know if you have any further questions.

**Common Concern 1:** Fairness and Budget Accounting: The paper’s comparison between LTS and DFS is weakened by inconsistent budget definition

**Response:**

Budget Measurement with Time-Based Units:
As we employ the same underlying tree-of-thoughts framework across all methods, we observed comparable tokens per thought. To address concerns regarding equitable budget allocation between DFS/Beam and LTS, we conducted additional experiments using wall-clock time for LM inference as the budget metric, rather than thought counts. The results and corresponding analysis are presented in the newly added Section 4.6 and Figure 4. Our findings demonstrate that LTS achieves comparable or superior accuracy to DFS across all three experimental settings under time-based budget constraints.
To contextualize the computational overhead, we measured inference times on PrOntoQA (averaged over 10 samples): thought generation required 0.67 ± 0.05 seconds per thought, while state evaluation required 5.88 ± 0.12 seconds. Following prior work (Hao et al., 2023), evaluation for PrOntoQA employed two sequential LM calls per thought (designated as "intuition" and "self-evaluation") that could not be parallelized due to GPU memory constraints. Additionally, the number of input tokens required for self-evaluation and intuition was approximately 1.8× that required for thought generation (i.e., the prompt size for PrOntoQA). In our initial query-based experiments (Section 4.5), we adopted a conservative accounting approach by treating each evaluation as a single unit to maintain consistency across domains. In contrast, the time-based experiments (Section 4.6) capture computational costs of all LM queries.

Section 4.6 (Section 4.7 in revised paper) results:
In the experiments presented in Section 4.6 (Section 4.7 in the revised paper), the computational budget was calibrated such that DFS successfully reached terminal states for all problem instances, whereas LTS did not always reach termination within the same budget. Despite this advantage, DFS exhibits lower or equivalent accuracy compared to LTS. This phenomenon exemplifies the second failure mode of DFS identified in Section 4.2.2: the initial-commitment problem, wherein DFS's greedy exploration strategy becomes trapped in locally optimal but globally suboptimal search trajectories. That is, providing additional budget to DFS does not address this limitation, as DFS returns a (non-goal) terminal node as answer. To avoid confusion we have clarified by adding the text “the budget was tuned such that DFS could reach a terminal state for every instance," as well as dropped the budgets column from the table.

Sort Domain Budget Accounting:
Regarding concerns about potential double-counting of computational costs for DFS in the Sort domain: our implementation leverages LM-assigned log probabilities directly as search heuristics, thereby obviating the requirement for auxiliary evaluation queries. Consequently, each thought incurs a single budget unit rather than two. This methodological choice is explicitly documented in Section 4.3: "For the Sort domain, we rely solely on the LM log probabilities to guide DFS. As a result, thoughts are not counted twice toward the budget in this setting, since no additional LM queries are required for state evaluation." This design ensures fair comparison by eliminating evaluation overhead that would disproportionately penalize DFS in domains where direct probability-based heuristics are viable.

**Common Concern 2:** Evaluations on other model family

**Response:** Given the concern on only using Llama models we provide additional results on another open source alternative Qwen 2 7B. The results follow a similar trend as for Llama, that is, LTS is on-par or better than DFS when budget is tuned so DFS can solve all instances (Section 4.6/7) (with 8.8% better accuracy for PrOntoQA). We only choose small models as, in this paper, we focus on LM inference on edge devices where compute is limited and using large models is infeasible. We have revised the text in Section 4.1 “smaller models allow us to rigorously assess test-time compute efficiency in compute-limited scenarios, which is a core focus of this work” as well as appended the results on Qwen 2 7B in Table 1.

---

> ### Author Response · Authors · 2025-11-26
> **Common Concern [2/2]**
>
> **Common Concern 3:** Sensitivity analysis of the proposed method
>
> **Response:** We thank the reviewer for this suggestion. We have added a temperature sensitivity analysis in Section 4.8 and Figure 5.
>
> We evaluated LTS across $\tau \in \{0.01, 0.5, 1.0, 1.5, 2.0\}$ on Blocksworld Step 4. The results reveal three behavioral regimes: (1) Low temperature (tau = 0.01) produces DFS-like behavior with good initial performance but early plateauing due to initial-commitment; (2) Intermediate temperatures (tau = 0.5-1.5) achieve optimal performance, showing continuous improvement and reaching the highest final accuracies; (3) High temperature (tau = 2.0) exhibits BFS-like behavior with poor initial performance but eventually might lead to competitive performance.
>
> These results suggest that temperature selection involves a trade-off between early performance and ultimate solution quality. In Blocksworld step 4 domain, $\tau \approx 0.5$-$1.0$ offers reasonable early performance while maintaining capacity for continued improvement under budget constraints.

---

### Author Response · Authors · 2025-12-18
**Following Up**

Dear Action Editor and Reviewers,

We wanted to inquire whether any additional information or clarification is needed from our side to assist with the review of the submission.

Thank you again for time and thoughtful reviews.

---

> ### Comment · Action_Editor_NGiv · 2025-12-19
>
> Congratulations to acceptance!
>
> Your Action Editor

---

> > ### Author Response · Authors · 2025-12-23
> > **Thank you!**
> >
> > Dear Action Editor and Reviewers,
> >
> > We would like to sincerely thank you for your time, effort, and constructive feedback throughout the review process. We truly appreciate your valuable comments and support.
> >
> > We have uploaded the camera-ready version of the paper. Please let us know if any further action is required from our side.
> >
> > Thank you once again for your efforts.

---

### Decision · Action_Editor_NGiv · 2025-12-16

**Recommendation:** Accept as is

**Additional Comments:**

Tree-of-thoughts (ToT) is a paradigm for exploring multiple reasoning paths in large language models (LLMs). This paper proposes using Levin tree search for choosing thoughts to expand in a ToT. The method is analyzed and its benefit is shown empirically on 3 problem classes. The paper was well received by the reviewers. The reviewers had three major concerns, which the authors addressed as follows:

* **Comparison to LTS and DFS:** Additional experiments with new budget metrics that are more favorable to LTS and DFS.

* **Evaluations on one model family:** Additional experiments with Qwen.

* **Sensitivity to temperature:** Additional experiments showing that the proposed method performs the best when the temperature is moderate.

All reviewers suggest accepting this paper and I support this decision.

**Audience:**

Yes

**Audience Explanation:**

Reasoning is a popular topic and doing it efficiently is an important research direction. I have no worries about audience for this paper.

**Claims And Evidence:**

Yes

**Claims Explanation:**

The paper proposes using Levin tree search for choosing thoughts to expand in a tree-of-thoughts (ToT). The method is analyzed and its benefit is shown empirically on 3 problem classes.